# Association of Serum High-Density Lipoprotein Cholesterol with High Blood Pressures at Checkup: Results of Kanagawa Investigation of Total Checkup Data from the National Database-9 (KITCHEN-9)

**DOI:** 10.3390/jcm10215118

**Published:** 2021-10-30

**Authors:** Kei Nakajima, Manami Igata, Ryoko Higuchi, Kotone Tanaka, Kaori Mizusawa, Teiji Nakamura

**Affiliations:** 1School of Nutrition and Dietetics, Faculty of Health and Social Services, Kanagawa University of Human Services, 1-10-1 Heisei-cho, Yokosuka 238-8522, Japan; 62020002.pn0@kuhs.ac.jp (M.I.); higuchi-nk3@kuhs.ac.jp (R.H.); tanaka-rt8@kuhs.ac.jp (K.T.); mizusawa.hsp@kuhs.ac.jp (K.M.); nakamura-t@kuhs.ac.jp (T.N.); 2Saitama Medical Center, Department of Endocrinology and Diabetes, Saitama Medical University, 1981 Kamoda, Kawagoe 350-8550, Japan

**Keywords:** high-density lipoprotein cholesterol, hypertension, blood pressure, low high-density lipoprotein cholesterol, extremely high high-density lipoprotein cholesterol, body mass index, big data

## Abstract

Background: although high-density lipoprotein has cardioprotective effects, the association between serum high-density lipoprotein cholesterol (HDL-C) and hypertension is poorly understood. Therefore, we investigated whether high and low concentrations of HDL-C are associated with high blood pressure (HBP) using a large healthcare dataset. Methods: in a community-based cross-sectional study of 1,493,152 Japanese people (830,669 men and 662,483 women) aged 40–74 years who underwent a health checkup, blood pressures automatically measured at healthcare center were investigated in nine HDL-C groups (20–110 mg/dL or over). Results: crude U-shaped relationship were observed between the nine HDL-C and blood pressures in both men and women. Logistic regression analysis showed left-to-right inverted J-shaped relationships between HDL-C and odds ratios for HBP (≥140/90 mmHg and/or pharmacotherapy), with lower limits of 90–99 mg/dL in both sexes, which were unchanged after adjusting for confounding factors. However, further adjustment for body mass index and serum triglyceride concentration revealed positive linear associations between HDL-C and HBP, although blunt U-shaped associations remained in nonalcohol drinkers. Conclusion: both low and extremely high HDL-C concentrations are associated with HBP. The former association might be dependent on excess fat mass concomitant with low HDL-C, whereas the latter association may be largely dependent on frequent alcohol consumption.

## 1. Introduction

High-density lipoprotein (HDL) is considered to have cardioprotective effects, which were confirmed repeatedly in molecular, cellular, animal, and human studies [1,2,3,4,5]. Therefore, individuals with a high serum HDL cholesterol (HDL-C) concentration are considered to be at a lower risk of cardiovascular disease (CVD) and mortality according to the concepts of “the higher, the better” and “longevity syndrome” [2,6,7]. However, in the past decade, several studies showed a U-shaped relationship between HDL-C and CVD, non-CVD, and all-cause mortality [8,9,10,11,12,13,14], which suggests that, like low HDL-C, very or extremely high HDL-C is not beneficial for overall health. Moreover, increasing HDL-C with pharmacotherapy, including cholesteryl ester transfer protein (CETP) inhibitors, did not show protective effects against CVD and mortality [15,16,17,18]. Among the plausible causes of adverse reactions with CETP inhibitors (torcetrapib and evacetrapib), a slight increase in blood pressure [19,20] and vasoactive effects [21] were observed and possibly were considered as markers of profound adverse reactions due to neuroendocrine or vasomotor effects [15].

Hypertension is a leading cause of CVD and mortality worldwide. However, to date, the fundamental relationship between serum HDL-C concentration and blood pressure was poorly argued, probably because an inverse relationship was observed between serum HDL-C and CVD and its risk factors [4,5].

Unexpectedly, however, instead of an inverse relationship, a slight U-shaped relationship, a left-to-right inverted J-shaped relationship [9,11,13], or a positive linear relationship [12,14] between circulating HDL-C and blood pressure or hypertension were observed in several studies. Despite these relationships being noted, authors did not particularly address them; hence, their statistical significance is unclear.

To date, three studies addressed the association between HDL-C concentration and hypertension [22,23,24]. Additionally, we recently showed an association between extremely high HDL-C (≥110 mg/dL) and hypertensive retinopathy in a general population who underwent fundus examinations [25]. Our results suggested a positive association between HDL-C concentration and blood pressure/hypertension-related pathology. However, these four studies [22,23,24,25], which examined the pathophysiology of very low and high HDL-C concentrations (e.g., <30 mg/dL and ≥90 mg/dL), consisted of relatively small sample sizes (<5000 participants in total). Actually, the number of subjects with hypertensive retinopathy was only 16 in the extremely high HDL-C group in our previous study [25]. Consequently, it was difficult to conduct an appropriate statistical analysis with proper adjustment for relevant confounding factors (covariates) in the low and very high HDL-C groups, which hampered disclosure of a latent overall association between HDL-C concentration and hypertension.

In this context, we investigated the crude and latent (adjusted for multiple background factors) associations between serum HDL-C concentration and high blood pressure (HBP) measured at checkup in a community-based cross-sectional study using a big healthcare dataset of 1.5 million general Japanese people [26]. Although hypertension is to be determined after consideration of home blood pressures [27,28], it is unfeasible to instruct the measurement of home blood pressure to huge population without suspicion for hypertension. Therefore, in this study, we investigated HBP at checkup, instead of hypertension.

## 2. Methods

### 2.1. Study Design and Subjects

We performed a composite multidisciplinary study involving secondary use of annual health checkup data in Japan (Kanagawa Investigation of the Total Checkup Data from the National Database [KITCHEN]) to investigate clinical factors primarily associated with cardiometabolic disease. Details of the study concept and design were published elsewhere [26]. Since 2008, all Japanese people aged 40–74 years are supposed to undergo a yearly itemized health checkup managed by the Ministry of Health, Labour, and Welfare (MHLW) [26,29]. The present study included all individuals who underwent these specific health checkups and who were living in Kanagawa Prefecture.

After religious review of our research project by the MHLW of Japan, our protocol was accepted in December 2016. We received digitally recorded anonymous data from the MHLW of Japan in September 2017, as part of its nationwide program involving the provision of medical data to third parties [30]. To conceal the identity of specific individuals, their ages were categorized as 40–44, 45–49, 50–54, 55–59, 60–64, 65–69, or 70–74 years. In this study, however, to evaluate subject age as a single numeric value, we transformed the age groups into substituted ages corresponding to the median of each age group (42, 47, 52, 57, 62, 67, and 72 years).

We initially reviewed data collected from 1,819,173 people aged 40–74 years who attended health checkups between April 2014 and March 2015. After excluding subjects with at least one missing continuous or categorical data point, 1,493,152 subjects were included in the study analysis (830,669 males and 662,483 females). Some proportions of subjects were treated with pharmacotherapy for hypertension, diabetes, and dyslipidemia. However, it is unclear whether individuals with treatment by a CEPT inhibitor were included in this study.

### 2.2. Measurements

Anthropometric and laboratory measurements were obtained on the morning following an overnight fast. Body weight and height were objectively measured by trained institutional staff members. Body mass index (BMI) was calculated as mass (in kg) divided by the square of height (m^2^). Biochemical measurements were performed automatically using standard methods. Serum low-density lipoprotein cholesterol (LDL-C), HDL-C, and triglyceride (TG) concentrations were measured automatically following rigorous instructions from the MHLW [26].

In a quiet room conditioned preferably at 20–25 °C, after five minutes of resting in the sitting position, blood pressure at the upper arm was determined using an automated upper arm cuff type (typically, 13 cm in width and 22–24 cm in length) sphygmomanometer at the healthcare institute performing the checkups [26,31,32]. Blood pressure was measured once in 70% of patients and twice in 30% of patients, who had suspicions about the first result of blood pressure due to the inadequate resting or measurements. The first result was used in 20% of subjects who confirmed that fist measurement was properly conducted after additional measurement. Otherwise, the second result was used in 10% of subjects.

HBP was defined as having either of systolic blood pressure of ≥140 mmHg, diastolic blood pressure of ≥90 mmHg, or pharmacotherapy for hypertension. Pulse pressure was calculated as systolic minus diastolic blood pressure. Subjects were classified into nine categories at 10-mg/dL HDL-C concentration intervals: 20–39, 40–49, 50–59, 60–69, 70–79, 80–89, 90–99, 100–109, and ≥110 mg/dL.

Because serum HDL-C concentration and blood pressure are elevated in people who frequently consume alcohol [33,34], to eliminate the effect of alcohol consumption on serum HDL-C and blood pressure, we investigated the association between these variables in a subgroup of subjects who answered “hardly drink (including cannot drink)” when asked “How often do you drink alcohol (sake, distilled spirits, beer, liquor and so)?” [26]. Other answers were “everyday” and “occasional”.

We also investigated the association between HDL-C concentrations and pharmacotherapy for hypertension, which was diagnosed considering home blood pressure independently of current blood pressure measured at the checkup. These results are provided in the Appendix A.

### 2.3. Statistical Analysis

Data are expressed as mean ± standard deviation or median (interquartile range). Differences in continuous and categorical variables were evaluated by analysis of variance (ANOVA) and the χ^2^ test, respectively. Trends in the prevalence of dichotomized categorical variables across the increasing HDL-C strata were evaluated by Cochran–Armitage tests. A logistic regression model was used to evaluate the associations between the nine HDL-C concentration categories and HBP with adjustment for potential confounding factors (age, sex, BMI, TG, LDL-C, glycated hemoglobin (HbA1c) (National Glycohemoglobin Standardization Program value), pharmacotherapy for diabetes mellitus, dyslipidemia, smoking, frequency of alcohol consumption, and habitual exercise), and yielded adjusted odds ratios (ORs) and 95% confidence intervals (CIs). Of note, because in general, HDL-C is inversely associated with BMI and serum TG concentration, the adjustment for BMI and TG was separately conducted. All statistical analyses were performed using SAS-Enterprise Guide (SAS-EG 7.1) in SAS software, version 9.4 (SAS Institute, Cary, NC, USA). *p* values of <0.05 were considered statistically significant.

## 3. Results

Table 1 shows the characteristics of subjects according to the nine HDL-C concentration categories. There were 50,064 subjects with an HDL-C concentration of ≥100 mg/dL (3.4% of the total), and nearly 75% of these were female. Overall, all continuous variables (blood pressure, pulse pressure, BMI, TG, LDL-C, and HbA1c), except age, were lower in the higher HDL-C groups compared with that of the lower HDL-C groups (ANOVA, all *p* < 0.0001). The prevalence of pharmacotherapy (for hypertension, diabetes mellitus, and dyslipidemia), CVD history, and current smoking were also lower (Cochran–Armitage test, all *p* < 0.0001). However, the prevalence of the female sex, daily alcohol consumption, and habitual exercise was higher in the higher HDL-C groups compared with that of the lower HDL-C groups (all *p* < 0.0001).

U or left-to-right inverted J-shaped relationships were observed between blood pressures and the nine HDL-C concentration categories in both men and women (Figure 1). Similar trends were also observed in the relationships between pulse pressures and the nine HDL-C concentrations.

HBP was observed in 31.7% of the whole, 36.4% of men, and 25.8% of women, respectively. Table 2 shows the prevalence of HBP in the nine HDL-C concentrations. Left-to-right inverted J-shaped relationships with a bottom of HDL-C of 90–99 mg/dL were observed between HBP and HDL-C categories in the whole (23.7%) and women (19.7%), whereas U-shaped relationship with a bottom of HDL-C of 70–79 mg/dL (33.8%) were observed in men. Of note, high prevalence of HBP was observed in both ends of HDL-C category in men (40.9% for the lowest and 40.8% for the highest).

In the results of logistic regression analysis, similar trends were observed between the nine HDL-C categories and their crude and confounders-adjusted ORs for HBP (Figure 2). Left-to-right inverted J-shaped relationship with a bottom of HDL-C of 90–99 mg/dL remained in men after adjustment for confounders except for BMI and serum TG. However, after further adjustment for BMI and serum TG, the associations between low and normal HDL-C categories (≤90mg/dL) and HBP were inverted (the ORs were reduced to less than 1.0) and the overall relationships between HDL-C concentration and HBP transformed into positive linear relationships in both men and women, which remained after adjustment for LDL-C and HbA1c.

Although data are not expressed, the daily and occasional alcohol consumptions were significantly associated with HBP compared with that of no alcohol consumption in the whole subjects (OR (95%IRs), 1.72 (1.71–1.74) and 1.59 (1.57–1.61)), even after adjustment for age, sex, smoking, habitual exercise, and pharmacotherapy for diabetes and dyslipidemia.

When the outcome of HBP was replaced with the presence of pharmacotherapy for hypertension in the same models above (Appendix A), left-to-right inverted J-shaped relationship with a bottom of HDL-C of 100–109 mg/dL were observed in the whole. However, after adjustment for confounders including BMI and TG, instead of positive linear, blunt U-shaped relationship with a bottom of HDL-C of 70–79 mg/dL for men and 80–89 mg/dL for women were observed.

### 3.1. Subgroup Analysis

In the subgroup of nondrinkers (n = 595,268), HBP was observed in 29.6% of the whole, 31.5% of men, and 28.4% of women, respectively. As shown in the Table 3, the prevalences of HBP were decreased across the increasing HDL-C in the whole subjects and women (*p* < 0.0001, Cochran–Armitage tests), whereas left-to-right inverted J-shaped relationship with a bottom of HDL-C of 90–99 mg/dL was observed in men.

Likewise, a crude logistic regression analysis showed inverse linear associations between HDL-C categories and HBP in the whole subjects and women, whereas left-to-right inverted J-shaped relationship were observed in men (Figure 3). These trends were not altered after adjustment for smoking; habitual exercise; and pharmacotherapy for dyslipidemia and diabetes. However, after further adjustment for BMI and serum TG, the positive associations between low and normal HDL-C categories (≤90 mg/dL in men) and HBP were attenuated but remain, and the relationships transformed into blunt U-shaped relationships with a bottom of HDL-C of 60–69 mg/dL in both men and women.

### 3.2. Discussion

Using a large healthcare dataset of 1.5 million general people, our study investigated complicated associations between serum HDL-C concentration and blood pressure, both of which are pivotal to the incidence of CVD and mortality and are simultaneously influenced by genetic background, comorbidities, and lifestyle factors, such as alcohol consumption, exercise, smoking, and body weight.

At first, our study demonstrated crude U-shaped relationship between serum HDL-C concentration and blood pressures. Similar relationships were also observed between serum HDL-C concentration and pulse pressure, which is a marker of atherosclerosis and a hyperdynamic circulation [35,36].

Overall, confounders adjusted ORs showed left-to-right inverted J-shaped relationship with a bottom of HDL-C of 90–99 mg/dL in men and women, which suggests that both low and extremely high HDL-C (≥100 mg/dL) are likely to be associated with HBP.

Positive association between extremely high HDL-C and HBP remained even after adjusting for BMI and serum TG in nondrinkers. Adjustment for BMI and serum TG may disclose different pathophysiology by eliminating the effect of excess body weight. Because BMI and serum TG were normal or low on average in the very and extremely high HDL-C groups (≥100 mg/dL), adjustment for BMI and TG did not change the association between high HDL-C concentration and HBP.

However, the positive association was attenuated even in men when subjects were restricted to the subgroup of nondrinkers, suggesting that this association may be largely dependent on frequent alcohol consumption, which was shown to raise serum HDL-C via inhibition of CETP activity and activation of lipoprotein lipase [37,38].

On the other hand, positive associations between low HDL-C categories and HBP were disappeared and inverted after adjustment for BMI and serum TG. Because a high BMI and serum TG concentration usually reflect obesity, positive associations between low HDL-C categories and HBP might be dependent on an excess body fat mass.

Intriguingly, such inversions were not observed in the subgroup of nondrinkers and when HBP was replaced with pharmacotherapy for hypertension. The former may be attributable to relatively lower BMI and serum TG in the nondrinkers compared with drinkers, which were confirmed in this study (albeit data is not shown), whereas the cause for the latter is unknown.

In this study, blood pressure was measured with an automated sphygmomanometer at checkup center but not at home. Therefore, the HBP determined in this study is likely to include the effect of white-coat hypertension [39,40]. However, as shown in Appendix A, similar U or J-shaped association of low and high HDL-C were observed with the pharmacotherapy for hypertension, which reflects already diagnosed hypertension in a clinical practice and is unrelated with the blood pressures measured at a checkup. In addition, current blood pressures are not conventional blood pressures measured manually in the presence of attending physician or nurse in a clinical practice. Furthermore, in recent years, it was shown that automatically measured office blood pressures are comparable to home blood pressures [41,42,43]. These may support current main results of relationship between HDL-C and HBP.

The underlying mechanism behind these fundamental positive linear associations is unknown. Unmeasured factors, such as heart rate; endocrine disorders accompanied by elevated catecholamines, cortisol, aldosterone, thyroid hormones, and sodium consumption, might contribute to these associations. In addition, it is unknown whether the nature of HBP in individuals with high HDL-C, if any, differs from that in individuals with low HDL-C, which needs further study.

In recent years, several studies showed that extremely high serum HDL-C may not be protective against CVD development [8,9,10,11,12,13,14]. Some of these studies presented U-shaped or positive linear relationships between HDL-C concentration categories and blood pressure [9,11,13], although the authors did not particularly argue them in their articles. Furthermore, none of these studies investigated pulse pressure in terms of HDL-C concentration categories.

In addition, clinical trials evaluating the effects of CETP inhibitors, which do not show a protective effect against CVD events, demonstrated a slightly increase in systolic blood pressure of 1.2–5.4 mmHg after intervention, concomitant with a substantial increase in HDL-C concentration [15,16].

Likewise, in our study, the difference in systolic blood pressure between the reference HDL-C concentration group and the extremely high HDL-C concentration group was small (6 mmHg at most in males) (Figure 1), which is coincidently equivalent to the increase in blood pressure caused by CETP inhibitors, as outlined above. However, it is unknown whether there is a common mechanism underpinning the pathophysiology between a naturally high HDL-C concentration and HDL-C raised by pharmacotherapy. Further study will be needed to confirm whether this small increase in blood pressure has clinical significance in patients with atherosclerosis and CVD.

### 3.3. Limitations

This study has some limitations. Firstly, strictly speaking, HBP at checkup is unlikely to be hypertension, which warrants further study including home blood pressures. Secondly, the causality between HDL-C concentration and HBP could not be identified, since this study adopted a cross-sectional design, despite of the large database. Thirdly, because subjects in this study were general people who underwent a health checkup, most were healthy and free from CVD and hypertension. Therefore, the results may not be applicable to other populations or patients with CVD and hypertension. Forthly, secondary hypertension and dyslipidemia, and genetic familial hypercholesterolemia were not excluded in this study, if any. Finally, the contents of pharmacotherapy for hypertension, which can influence the metabolism of lipids [44,45], the data of renal function and serum uric acid, which contribute to the development of hypertension [46,47,48,49], and the data of diet were also unavailable in this study. Further studies addressing these limitations are needed to warrant current results.

## 4. Conclusions

The association between HDL-C concentration and blood pressure is complicated and differs between people who consume alcohol and those who do not. In clinical practice, an apparent inverse association between HDL-C concentration (30–70 mg/dL, which is the frequently observed range) and HBP may be observed, although this association can depend on body fat mass. The robust association between an extremely high HDL-C concentration and HBP, particularly in men, and the fundamental positive linear association between HDL-C concentration and HBP may challenge the traditional concept of “the higher, the better” and the high HDL-C concentration caused by CETP inhibitors. Further studies, including long-term prospective studies and clinical trials, on pharmacotherapy intervention are needed to confirm our results.

## Figures and Tables

**Figure 1 jcm-10-05118-f001:**
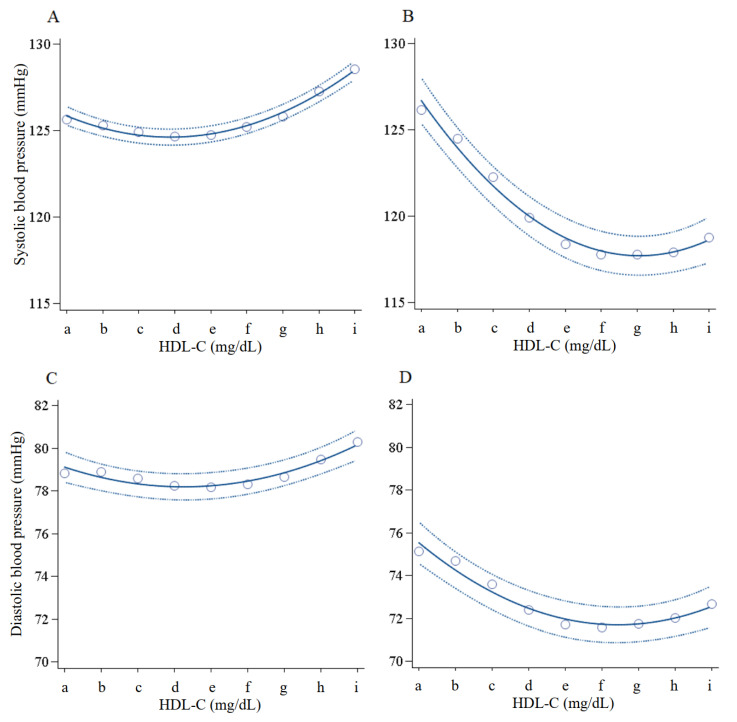
Blood pressure in men and women according to nine HDL-C concentration categories. With HDL-C concentrations of 20–39, 40–49, 50–59, 60–69, 70–79, 80–89, 90–99, 100–109, 110–119, and ≥120 mg/dL, n = 57,052, 193,722, 235,749, 171,612, 95,359, 45,195, 19,204, 7624, and 5152 for men and 6800, 44,379, 112,421, 160,350, 148,963, 99,424, 52,858, 22,886, and 14,402 for women, respectively. Open circle expresses mean of blood pressure in each HDL-C concentration category. Solid line and dashed line express quadratic regression curve and 95%CIs. a, 20–39; b, 40–49; c, 50–59; d, 60–69; e, 70–79; f, 80–89; g, 90–99; h, 100–109; i, ≥110 mg/dL. (**A**): Systolic blood pressure in men; (**B**): Systolic blood pressure in women; (**C**): Diastolic blood pressure in men; (**D**): Diastolic blood pressure in women; (**E**): Pulse pressure in men; (**F**): Pulse pressure in women.

**Figure 2 jcm-10-05118-f002:**
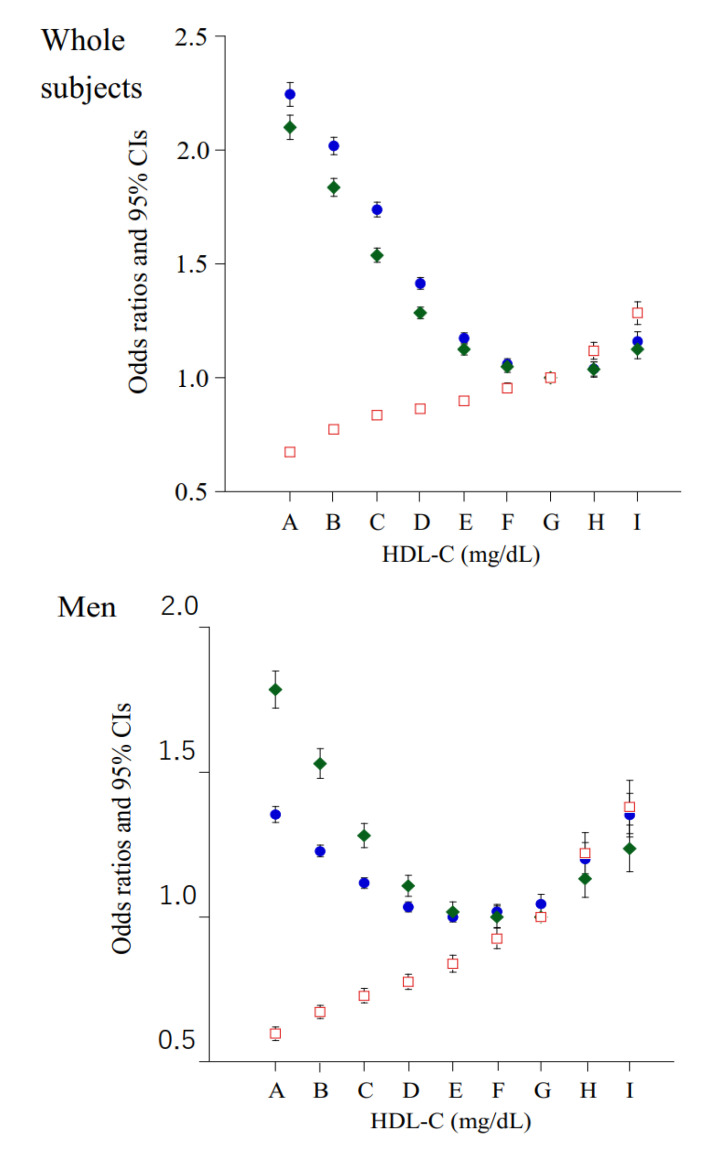
Odds ratios of nine HDL-C concentration categories for HBP. Closed blue circle: unadjusted. Closed green diamond: adjusted for age, sex (whole subjects), smoking, pharmacotherapy (for diabetes mellitus or dyslipidemia), habitual exercise, and daily alcohol consumption. Open red square: plus adjusted for body mass index and serum concentration of triglyceride A, 20–39; B, 40–49; C, 50–59; D, 60–69; E, 70–79; F, 80–89; G, 90–99; H, 100–109; I, ≥110 mg/dL.

**Figure 3 jcm-10-05118-f003:**
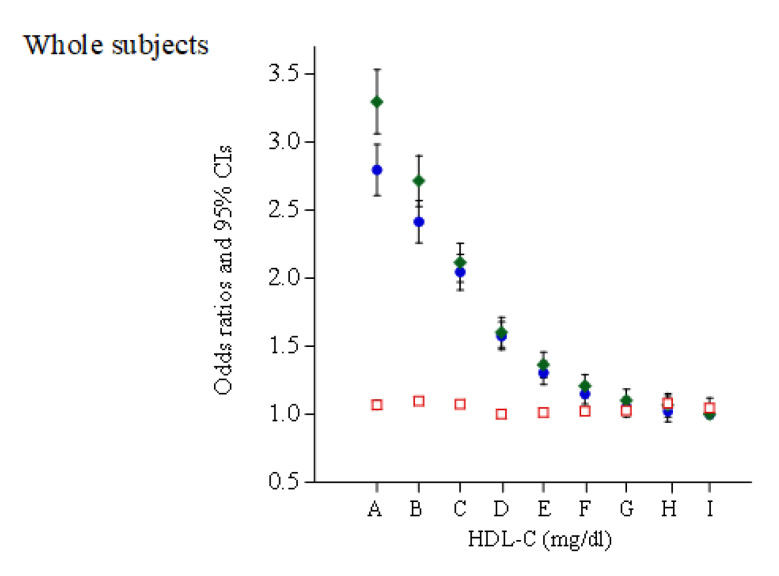
Odds ratios of nine HDL-C concentration categories for HBP in subjects who hardly consumed alcohol. Closed blue circle: unadjusted. Closed green diamond: adjusted for age, sex (whole subjects), smoking, pharmacotherapy (for diabetes mellitus or dyslipidemia), habitual exercise, and daily alcohol consumption. Open red square: plus adjusted for body mass index and serum concentration of triglyceride A, 20–39; B, 40–49; C, 50–59; D, 60–69; E, 70–79; F, 80–89; G, 90–99; H, 100–109; I, ≥110 mg/dL.

**Table 1 jcm-10-05118-t001:** Clinical characteristics of subject.

HDL-C Categories (mg/dL)	30–39	40–49	50–59	60–69	70–79	80–89	90–99	100–109	≥110
N (% of total)	63,852 (4.3)	238,101 (16.0)	348,170 (23.3)	331,962 (22.2)	244,322 (16.4)	144,619 (9.7)	72,062 (4.8)	30,510 (2.0)	19,554 (1.3)
s-Age	54.0 ± 10.0	54.2 ± 10.0	54.8 ± 10.1	55.1 ± 10.2	55.1 ± 10.1	55.2 ± 10.0	55.4 ± 9.7	55.6 ± 9.4	55.8 ± 9.2
Women, n (%)	6800 (10.7)	44,379 (18.6)	112,421 (32.3)	160,350 (48.3)	148,963 (61.0)	99,424 (68.8)	52,858 (73.4)	22,886 (75.0)	14,402 (73.7)
BMI (kg/m^2^)	25.9 ± 3.7	25.0 ± 3.6	23.9 ± 3.4	22.8 ± 3.2	21.8 ± 3.0	21.1 ± 2.8	20.7 ± 2.7	20.4 ± 2.6	20.1 ± 2.6
SBP (mmHg)	126 ± 16.4	125 ± 16.4	124 ± 16.8	122 ± 17.2	121 ± 17.5	120 ± 17.6	120 ± 17.6	120 ± 17.9	121 ± 18.2
DBP (mmHg)	78.4 ± 11.5	78.1 ± 11.5	77.0 ± 11.6	75.4 ± 11.7	74.3 ± 11.7	73.7 ± 11.7	73.6 ± 11.6	73.9 ± 11.8	74.7 ± 12.0
PP (mmHg)	47.3 ± 11.1	47.0 ± 11.0	47.1 ± 11.2	47.0 ± 11.3	46.6 ± 11.4	46.4 ± 11.4	46.3 ± 11.4	46.4 ± 11.5	46.7 ± 11.6
Triglyceride (mg/dL)	185 (130–270)	138 (99–193)	106 (77–146)	85 (64–117)	73 (56–98)	66 (51–87)	62 (49–81)	59 (47–76)	58 (46–75)
LDL-C (mg/dL) *	121 ± 33.4	130 ± 32.4	130 ± 32.2	126 ± 31.6	122 ± 30.8	120 ± 30.2	118 ± 30.3	117 ± 30.8	112 ± 32.6
HDL-C (mg/dL)	36.0 ± 2.9	45.1 ± 2.8	54.6 ± 2.9	64.3 ± 2.9	74.2 ± 2.8	84.0 ± 2.8	93.8 ± 2.8	103.8 ± 2.8	120.4 ± 11.9
HbA1c (NGSP, %) **	5.9 ± 0.9	5.7 ± 0.8	5.6 ± 0.7	5.5 ± 0.5	5.5 ± 0.5	5.5 ± 0.4	5.4 ± 0.4	5.4 ± 0.4	5.4 ± 0.5
Pharmacotherapy for									
hypertension, n (%)	16,700 (26.2)	56,096 (23.6)	73,636 (21.2)	59,303 (17.9)	36,521 (15.0)	19,188 (13.3)	8,810 (12.2)	3,677 (12.1)	2,533 (13.0)
diabetes, n (%)	5974 (9.4)	16,629 (7.0)	17,094 (4.9)	11,105 (3.4)	5720 (2.3)	2675 (1.9)	1123 (1.6)	490 (1.6)	343 (1.8)
dyslipidemia, n (%)	91,03 (14.3)	34,431 (14.5)	49,956 (14.4)	42,471 (12.8)	26,229 (10.7)	13,417 (9.3)	6005 (8.3)	2345 (7.7)	1410 (7.2)
Cardiovascular disease, n (%)	3,293 (5.2)	9,747 (4.1)	12,237 (3.5)	9,867 (3.0)	6,100 (2.5)	3,327 (2.3)	1,532 (2.1)	597 (2.0)	349 (1.8)
Current smokers, n (%)	28,658 (44.9)	81,543 (34.3)	89,236 (25.6)	63,958 (19.3)	37,275 (15.3)	18,864 (13.0)	8,550 (11.9)	3,667 (12.0)	2,629 (13.4)
Alcohol drinkers									
Everyday, n (%)	11,580 (18.1)	55,642 (23.4)	96,004 (27.6)	96,149 (29.0)	72,192 (29.6)	44,524 (30.8)	23,825 (33.1)	11,156 (36.6)	8,577 (43.9)
Occasional, n (%)	21,514 (33.7)	82,110 (34.5)	114,385 (32.9)	103,970 (31.3)	75,479 (30.9)	44,631 (30.9)	21,864 (30.3)	8,957 (29.4)	5,325 (27.2)
Non, n (%)	30,758 (48.2)	100,349 (42.2)	13,7781 (39.6)	131,843 (39.7)	96,651 (39.6)	55,464 (38.4)	26,373 (36.6)	10,397 (34.1)	5,652 (28.9)
Regular exercisers, n (%) ***	14,582 (22.8)	60,875 (25.6)	99,252 (28.5)	102,162 (30.8)	78,890 (32.3)	48,424 (33.5)	25,094 (34.8)	10,874 (35.6)	7,300 (37.3)

Data are presented as mean ± standard deviation, median (interquartile range) [triglyceride], or n (%). * Available n = 1,448,907. ** Available n = 1,260,117. *** Regular exercise defined as ≥30 min at least twice a week. All continuous and categorical variables show significant differences with an ANOVA or χ^2^ test, with *p* < 0.0001 across the nine HDL-C concentration categories. s-Age, substituted age; BMI, body mass index; HDL-C, high-density lipoprotein cholesterol; SBP, systolic blood pressure; DBP, diastolic blood pressure; PP, pulse pressure.

**Table 2 jcm-10-05118-t002:** Prevalence of HBP according to nine HDL-C concentration categories.

HDL-C Category (mg/dL)	20–39	40–49	50–59	60–69	70–79	80–89	90–99	100–109	≥110
Whole subjects (n = 1,493,152)									
Case of HBP, n (% in each group)	26,275 (41.2)	91,895 (38.6)	122,287 (35.1)	101,501 (30.6)	65,474 (26.8)	35,924 (24.8)	17,111 (23.7)	7461 (24.5)	5190 (26.5)
Men (n = 830,669)									
Case of HBP, n (% in each group)	23,324 (40.9)	74,652 (38.5)	85,647 (36.3)	59,333 (34.6)	32,232 (33.8)	15,473 (34.2)	6680 (34.8)	2896 (38.0)	2104 (40.8)
Women (n = 662,483)									
Case of HBP, n (% in each group)	2951 (43.4)	17,243 (38.9)	36,640 (32.6)	42,168 (26.3)	33,242 (22.3)	20,451 (20.6)	10,431 (19.7)	4565 (20.0)	3086 (21.4)

HBP was defined as blood pressure of ≥140/90 mmHg and/or pharmacotherapy for hypertension.

**Table 3 jcm-10-05118-t003:** Prevalence of HBP according to nine HDL-C concentration categories in subjects who hardly consumed alcohol.

HDL-C Category (mg/dL)	30–39	40–49	50–59	60–69	70–79	80–89	90–99	100–109	≥110
Whole subjects, n (% in total n = 595,268)	30,758 (5.2)	100,349 (16.9)	137,781 (23.1)	131,843 (22.1)	96,651 (16.2)	55,464 (9.3)	26,373 (4.4)	10,397 (1.7)	5,652 (0.9)
Case of HBP, n (% in each group)	12,465 (40.5)	37,204 (37.1)	45,813 (33.3)	36,585 (27.8)	23,333 (24.1)	12,139 (21.9)	5,379 (20.4)	2,078 (20.0)	1,108 (19.6)
Men, n (% in total n = 227,709)	25,865 (11.4)	70,200 (30.8)	66,599 (29.2)	38,101 (16.7)	16,939 (7.4)	6,499 (2.9)	2,345 (1.0)	755 (0.3)	406 (0.2)
Case of HBP, n (% in each group)	10,235 (39.6)	24,756 (35.3)	20,628 (31.0)	9,887 (26.0)	4,053 (23.9)	1,410 (21.7)	493 (21.0)	174 (23.1)	99 (24.4)
Women, n (% in total n = 367,559)	4,893 (1.3)	30,149 (8.2)	71,182 (19.4)	93,742 (25.5)	79,712 (21.7)	48,965 (13.3)	24,028 (6.5)	9,642 (2.6)	5,246 (1.4)
Case of HBP, n (% in each group)	2,230 (45.6)	12,448 (41.3)	25,185 (35.4)	26,698 (28.5)	19,280 (24.2)	10,729 (21.9)	4,886 (20.3)	1,904 (19.8)	1,009 (19.2)

HBP was defined as blood pressure of ≥140/90 mmHg and/or pharmacotherapy for hypertension.

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
