# Peer review of "Association of Serum High-Density Lipoprotein Cholesterol with High Blood Pressures at Checkup: Results of Kanagawa Investigation of Total Checkup Data from the National Database-9 (KITCHEN-9)"

_jcm, 2021, doi:10.3390/jcm10215118_

Round 1

Reviewer 1 Report

Prof Nakajima and colleagues present an observational study of nearly 1.5 million Japanese individuals aged 40-74 years, who underwent health check-ups, and investigated the relationship between high-density lipoprotein cholestrol concentration (HDL-C) with blood pressure measurements. The main finding is that there is a U-shaped association between the two, and that excess fat may explain the increased high blood pressure risk in the low HDL-C groups, and excess alcohol may explain the increased risk for the high HDL-C groups.

The study is interesting and utilises a large dataset indeed. The manuscript is well written, although can benefit from being more succinct at times. I enclose a few points for the authors' attention:

Introduction: is rather lengthy and can/should be made more succinct please

Methods: in patients with 2 BP measurements, how did the investigators decide which BP reading to take? I note that 20% had the first, and 10% had the second BP reading. Please clarify

Methods: the endpoint 'pharmacotherapy for hypertension' is too ambiguous. What medications were considered in this category? and what about medications that can be used for hypertension as well as other indications? (e.g. beta-blockers)

Methods: why did the authors 'bin' continuous variables into ordinal? this practice is strongly discouraged as it causes loss of power and accuracy (doi.org/10.1186/1471-2288-12-21 and https://doi.org/10.1097/00001648-199507000-00025). Also, the authors categorised the HDL-C concentrations in 5 categories in their recent hypertensive retinopathy paper, but opted for 9 in this paper. Can they please clarify?

Methods: how did the authors decide on what variables to include in their multiple models? and how did they decide at what time was a variable to be inserted into a particular model? I struggle to see the need for so many models and I found it quite hard to keep up with the results of all of these models

Figures: please consider providing the 95% confidence intervals pictorially 

Tables: please consider presenting the findings of table 2 as a figure.  In its current form, the table is way too busy and the reader would struggle to appreciate the described U and J relationships without graphical presentation

Author Response

Introduction: is rather lengthy and can/should be made more succinct please

Response:

We agree with this comment. We deleted some sentences from the introduction section.

Methods: in patients with 2 BP measurements, how did the investigators decide which BP reading to take? I note that 20% had the first, and 10% had the second BP reading. Please clarify

Response:

We agree with this comment. We added next sentences in the methods section (Line 143).

“Blood pressure was measured once in 70% of patients and twice in 30% of patients, who had suspicions about the first result of blood pressure due to the inadequate resting or measurements. The first result was used in 20% of subjects who confirmed that fist measurement was properly conducted after additional measurement. Otherwise, the second result was used in 10% of subjects.”

Methods: the endpoint 'pharmacotherapy for hypertension' is too ambiguous. What medications were considered in this category? and what about medications that can be used for hypertension as well as other indications? (e.g. beta-blockers)

Response:

We agree with this comment that alternative endpoint 'pharmacotherapy for hypertension' is too ambiguous. Therefore, this result is provided in a form of supplementary Figure. In addition, details of pharmacotherapy are unavailable, which has been described in the limitation section (Line 309).

Methods: why did the authors 'bin' continuous variables into ordinal? this practice is strongly discouraged as it causes loss of power and accuracy (doi.org/10.1186/1471-2288-12-21 and https://doi.org/10.1097/00001648-199507000-00025). Also, the authors categorised the HDL-C concentrations in 5 categories in their recent hypertensive retinopathy paper, but opted for 9 in this paper. Can they please clarify?

Response:

We have read the articles recommended and partially agree with the comment.

However, in our study, we categorized subjects into 9 groups based on every 10 mg/dl interval of serum HDL-C, but not on the data used for the study, which enables investigators to compare studies in the future. In addition, the heterogeneous in each category may be low because the range of 10 mg/dl is narrow.

In our recent hypertensive retinopathy paper, we had to categorize into five or less groups because the whole sample size was 4,072, which is very small for the investigation of extremely high HDL-C.

Methods: how did the authors decide on what variables to include in their multiple models? and how did they decide at what time was a variable to be inserted into a particular model? I struggle to see the need for so many models and I found it quite hard to keep up with the results of all of these models

Response:

We agree with this comment. We decided to add confounding variables based on the established clinical parameters (age, sex, smoking, exercise,..) but not computer selection such as stepwise regression method, because this method enables investigators to compare our study and other studies in the future.

In the revised version, we simplified and restricted to three models: unadjusted (crude), adjusted for clinical factors, and further adjustment for BMI and TG.

Figures: please consider providing the 95% confidence intervals pictorially 

Response:

We agree with this comment. We made a new figure with regression curve and 95% confidence intervals.

Tables: please consider presenting the findings of table 2 as a figure.  In its current form, the table is way too busy and the reader would struggle to appreciate the described U and J relationships without graphical presentation

Response:

We agree with this comment. We made three figures in place of tables.

Reviewer 2 Report

This study by Nakajima et al., investigated the association of serum high-density lipoprotein cholesterol with high blood pressures at checkup: results of Kanagawa Investigation of Total Checkup Data from the National Database-9 (KITCHEN-9). The aim of this study was an important area and well-designed study. The reviewer has some minor concerns,

  1. Authors should consider the model of diet intake low fat diet Vs high fat diet model if the data is available.
  2. Did the authors consider the genetic familial hypercholesterolemia role in the rise of blood pressure?
  3. It is not clear, the subjects included in the study were using CEPT inhibitors or other drugs that control hyperlipidemia, clarify this in the methods section.

Author Response

Authors should consider the model of diet intake low fat diet Vs high fat diet model if the data is available.

Response:

We agree with this comment. Unfortunately, we have no data of diet intake. Therefore, we added this in the limitation section (Line 311).

Did the authors consider the genetic familial hypercholesterolemia role in the rise of blood pressure?

Response:

Unfortunately, we have no data of familial hypercholesterolemia. Therefore, we added this in the limitation section (Line 308).

It is not clear, the subjects included in the study were using CEPT inhibitors or other drugs that control hyperlipidemia, clarify this in the methods section.

Response:

We agree with this comment. Unfortunately, we have no data of CEPT inhibitors. Therefore, we added this in the methods section as follows (Line 127).

“Some proportions of subjects had been treated with pharmacotherapy for hypertension, diabetes, and dyslipidemia. However, it is unclear whether individuals with treatment by a CEPT inhibitor were included in this study.”

Reviewer 3 Report

Association of high and low serum high-density lipoprotein cholesterol with hypertension: 1 results of Kanagawa Investigation of Total Checkup Data from the National Database-9 2 (KITCHEN-9)

Article is interesting.

In this study there were investigated whether high and low concentrations of HDL-C are associated with hypertension. The population under study is very large and fully representative and uses a proper methodology.

I wonder if such a conclusion “Both low and extremely high HDL-C concentrations are associated with hypertension” can be drawn using the current results. In Table 2 - Odds ratios of nine HDL-C concentration categories for hypertension was shown. Based on the results from Table 2, it can be concluded that the concentration of HDL-CH if it is lower than 90-99 mg/dL in all subgroups 20–39, 40–49, 50–59, 60–69, 70–79, 80–89 increases the risk of hypertension (model 1-3), as well as above 100 mg/dL. The authors did not prove that low HDL-CH concentration is associated with a greater prevalence of arterial hypertension. To demonstrate this, it would be necessary to additionally divide the study population into 3 groups - low, normal and high HDL-CH and calculate the OR.

Author Response

I wonder if such a conclusion “Both low and extremely high HDL-C concentrations are associated with hypertension” can be drawn using the current results. In Table 2 - Odds ratios of nine HDL-C concentration categories for hypertension was shown. Based on the results from Table 2, it can be concluded that the concentration of HDL-CH if it is lower than 90-99 mg/dL in all subgroups 20–39, 40–49, 50–59, 60–69, 70–79, 80–89 increases the risk of hypertension (model 1-3), as well as above 100 mg/dL. The authors did not prove that low HDL-CH concentration is associated with a greater prevalence of arterial hypertension. To demonstrate this, it would be necessary to additionally divide the study population into 3 groups - low, normal and high HDL-CH and calculate the OR.

Response:

We agree with this comment. In the whole subjects, we calculated OR for high blood pressure after division of subjects into 3 groups: low (20-89 mg/dl), normal(90-99mg/dl) and high HDL-C (> 100mg/dl).

The results ORs are 1.33 (1.31-1.36) (p<0.0001), 1.0 (reference), and 1.08 (1.05-1.11) (p<0.0001) controlling for age, sex, smoking, exercise, and pharmacotherapy for diabetes and dyslipidemia.

After further adjustment for BMI and TG, the results ORs are 0.88 (0.86-0.90) (p<0.0001), 1.0 (reference), and 1.18 (1.14-1.21) (p<0.0001).

These results suggest that “Both low and extremely high HDL-C concentrations are associated with hypertension”.

However, as the reviewer 1 suggested, tertile or quartile may sometimes cause a problem due to the heterogeneous in the category with a wide range (for instance, 20-89mg/dl). Therefore, we did not add the results in the manuscript.

Round 2

Reviewer 1 Report

Thank you for addressing the concerns raised in the previous review round. I have no further comments. 

Reviewer 3 Report

Perfect results.

ORs 1.33 (1.31-1.36) (p<0.0001), 1.0 (reference), and 1.08 (1.05-1.11) (p<0.0001) controlling for age, sex, smoking, exercise, and pharmacotherapy for diabetes and dyslipidemia. These results suggest exactly that “Both low and extremely high HDL-C concentrations are associated with hypertension”. It is a simple analysis, but corresponds very closely to the conclusions. 

The reader will judge the limitations of the reference group by himself.

Please attach this information to the manuscript.

This manuscript is a resubmission of an earlier submission. The following is a list of the peer review reports and author responses from that submission.

Round 1

Reviewer 1 Report

Dr Nakagima and colleagues present their analysis of the association between HDL-C and blood pressure using a large registry dataset. They observe a U-shaped relationship in men, and a left-to-right-inverted J relationship in women, challenging the concept of a linear relationship and limitless beneficial effect with higher HDL-C concentrations. This relationship remained true after adjusting for multiple covariates.

The study is well conceived and well written, for which the authors ought to be commended. I enjoyed reading the paper and I have no major comments. I would however have expected the graphs to have confidence intervals to be visually represented in the graphs.

Author Response

Q1: I would however have expected the graphs to have confidence intervals to be visually represented in the graphs.

Thank you for the comments.

Answer: We agree with this comment. However, the standard deviation is large particularly in the both ends of HDL-C groups (10-25 mmHg) for the figure, whereas the standard error and 95%CIs is very small (0.03-0.15mmHg). In this context, we did not add the SD, SE, and 95%CI. Because the readers may feel the same thing, we add the next sentence in line 201.

“Because standard error and 95%CIs for each point were very small (0.03-0.15 mmHg), these bars were omitted from the figures.”

Reviewer 2 Report

The topic is important and of interest in that the epidemiology of HDL-C is not entirely c/w clinical trial data; however, concerns exist:

1. Are methodological differences for lipoprotein measurements important?
2. Only 1 or 2 BP determinations were done and what determined which value was used when 2 were done?  This is an inadequate assessment of BP.
3. BP meds are important but adjustment relates only to presence or absence? 
4. Similar concern relates to adjustments for diabetes and lipid meds.

Author Response

Thank you for the comments.

Q1. Are methodological differences for lipoprotein measurements important?

A1: We believe that methodological differences for lipoprotein measurements are not important.

The sentence of “mainly spectrophotometrically (in ~85% of samples) using a direct, non-precipitation method, and the remainder were measured using other methods” may be confusing for the readers. Therefore, we deleted the sentence, and add “using a standard measuring equipment” instead in Line 140.

Q2: Only 1 or 2 BP determinations were done and what determined which value was used when 2 were done?  This is an inadequate assessment of BP.

A2: We agree with the comments. The reason why some proportion of subjects had BP measured twice was that they thought the first measurement was inadequate due to inadequate resting and procedures for the measurement. However, after two times measurements, most of them (20%/30%) believed the first measurement was not inadequate and then first BP was used. On the other hand, one third of subjects confirmed the measurement was inadequate and second measurement was used for the BP.

We changed the sentence as follows:

“Of subjects who underwent two blood pressure measurements due to the possibility for inadequate resting or measurements, the first result was used in 20% of subjects who confirmed that fist measurement was properly conducted after two measurements, while the second result was used in 10% of subjects.”

Because the term of patients is not appropriate, we changed them into “subjects.”

Q3: BP meds are important but adjustment relates only to presence or absence?

A3: We agree with this comment. Unfortunately, we had no data for the contents of BP medication.

We acknowledged that this is one of limitations in this study. Therefore, we added the next sentences in the limitation section.

Line 315,

Third, data on sodium consumption and other nutrients, including potassium and fiber, and the menopausal state in women (postmenopausal or not), and contents of medications for hypertension, diabetes, and dyslipidemia were unavailable in this study.

Q4. Similar concern relates to adjustments for diabetes and lipid meds.

A4: We agree with this comment. Unfortunately, we had no data for the contents of medications for diabetes and dyslipidemia. Therefore, likewise, we added the next sentences in the limitation section.

Line 315,

Third, data on sodium consumption and other nutrients, including potassium and fiber, and the menopausal state in women (postmenopausal or not), and contents of medications for hypertension, diabetes, and dyslipidemia were unavailable in this study.

Round 2

Reviewer 2 Report

The concerns are major and mostly remain.